# Nanoparticle-Induced m^6^A RNA Modification: Detection Methods, Mechanisms and Applications

**DOI:** 10.3390/nano12030389

**Published:** 2022-01-25

**Authors:** Yi Wang, Fengkai Ruan, Zhenghong Zuo, Chengyong He

**Affiliations:** State Key Laboratory of Cellular Stress Biology, School of Life Sciences, Shenzhen Research Institute of Xiamen University, Xiamen University, Xiamen 361005, China; 21620182203493@stu.xmu.edu.cn (Y.W.); 21620200156496@stu.xmu.edu.cn (F.R.); zuozhenghong@xmu.edu.cn (Z.Z.)

**Keywords:** nanosafety, (N^6^-methyladenosine) RNA modification, function of m^6^A, epigenetics

## Abstract

With the increasing application of nanoparticles (NPs) in medical and consumer applications, it is necessary to ensure their safety. As m^6^A (N^6^-methyladenosine) RNA modification is one of the most prevalent RNA modifications involved in many diseases and essential biological processes, the relationship between nanoparticles and m^6^A RNA modification for the modulation of these events has attracted substantial research interest. However, there is limited knowledge regarding the relationship between nanoparticles and m^6^A RNA modification, but evidence is beginning to emerge. Therefore, a summary of these aspects from current research on nanoparticle-induced m^6^A RNA modification is timely and significant. In this review, we highlight the roles of m^6^A RNA modification in the bioimpacts of nanoparticles and thus elaborate on the mechanisms of nanoparticle-induced m^6^A RNA modification. We also summarize the dynamic regulation and biofunctions of m^6^A RNA modification. Moreover, we emphasize recent advances in the application perspective of nanoparticle-induced m^6^A RNA modification in medication and toxicity of nanoparticles to provide a potential method to facilitate the design of nanoparticles by deliberately tuning m^6^A RNA modification.

## 1. Introduction

As innovative nanotechnology has developed, nanoparticles (NPs), sized between 1 and 110 nm, have been created to improve the quality of life and are widely applied, including in household items, building materials, food products, cosmetic products, biomedical imaging, biomedical diagnostics, drug delivery and anticancer therapy [1,2,3]. For example, silica dioxide nanoparticles (SiO_2_-NPs) [4] and titanium dioxide nanoparticles (TiO_2_-NPs) [5] have the ability to self-clean, repel water and resist heat, and they can be used as coatings. Nanoparticles coated with modified polyethylene glycol (PEG) through chemical bonds can function as drug carriers for oral peptide drugs, as they are more stable in gastrointestinal fluid and improve peptide absorption [6,7]. Quantum dots (QDs) are specific fluorescent nanomaterials that can be added to proteins to serve as molecular biomarkers of diseases and as tools for living imaging [8]. Nanoparticles have been used in diagnosis, imaging and drug delivery regarding inflammatory bowel disease (IBD) [9]. Additionally, nanoparticles were modified for possible anticancer application [10].

Because of the widespread use of NPs, their biological effects on organisms and the mechanisms involved have been extensively studied over the past few decades using many models, both in vitro and in vivo [11,12,13,14,15]. On the one hand, NPs have potential toxicity, and constant exposure to NPs has increased risks of neurodegeneration, immunological diseases and even cancers [16,17,18]. It has also been shown through the interactions between NPs and biomacromolecules, such as proteins, lipids and DNA, that NPs can cause inflammation, lipid peroxidation and oxidative stress [19,20,21,22]. Increasing evidence suggests that DNA methylation, miRNA regulation, histone modification, RNA modification and other epigenetic modulations play a significant role in the biological activity of NPs [23,24,25]. An increasing amount of research suggests that environmentally induced epigenetic modifications are involved in the initiation and processing of various diseases [26], such as obesity, dysplasia and even cancer, implying that NPs may have long-term negative health consequences. Many studies have been performed, but unfortunately, there is limited knowledge regarding the relationship between nanoparticles and m^6^A RNA modification, which is one of the most abundant mRNA modifications.

In this review, we provide a comprehensive overview of recent research progress on the dynamic regulation and bioimpacts of m^6^A RNA modification as well as its relationship to NP-induced bioimpacts, and we discuss future research directions with the goal of providing a theoretical foundation based on currently available literature for further investigating the relationship between m^6^A RNA modification and nanoparticle bioimpacts.

## 2. m^6^A RNA Modification

There have been more than 150 various RNA modifications reported in different organisms; since the first modification was found in 1957, the most abundant modification is the m^6^A RNA modification [27]. As the first RNA modification to have its “writers”, “erasers” and “readers” characterized, knowledge of its regulation and biofunctions is of great importance to better understand RNA modification. This modification is associated with the fate of mRNA, indicating that it may have an essential role in exact posttranscriptional regulation during gene expression. The correlation between m^6^A RNA modification and diseases indicates its potential application in toxicity- and medication-involved NPs.

### 2.1. Dynamic Regulation of m^6^A RNA Modification

Due to the shortage of proteins related to m^6^A and the short half-life of most types of RNA, m^6^A RNA modifications have been considered to be stable and unchanged for a long time. The discovery of the first authentic m^6^A demethylase, namely, fat mass and obesity associated protein (FTO) [28], resolved this conundrum, demonstrating that m^6^A RNA modification is a dynamic process that is tightly regulated. As more studies have been performed, it is known that m^6^A RNA modification is involved in “writers”, “erasers” and “readers”, which is a dynamic process to add or remove a methyl group to adenosine and usually occurs in the RRACH sequence of adenines. To visualize the location and distribution of these proteins involved in m^6^A, we outlined the types of proteins involved in m^6^A RNA modification (Figure 1).

#### 2.1.1. m^6^A “Writers”—Adenosine Methyltransferases

m^6^A RNA modification is modified by “writers”, including adenosine methyltransferases, methyltransferase-like 3 (METTL3), methyltransferase-like 14 (METTL14), Wilms tumour 1-associated protein (WTAP), KIAA1429 and RNA-binding motif protein 15/15B (RBM15/RBM15B). Well-known methyltransferases can add a methyl group to adenosine. The METTL3-METTL14-WTAP methyltransferase complex can add m^6^A, and METTL3 has been previously discovered to be an S-adenosyl-L-methionine (SAM)-binding site in this multiprotein complex, which is widely expressed and highly conserved. Through knockdown of *Mettl3* in blastocysts achieved by mating *Mettl3*^+/−^ mice, it has been shown that METTL3 is a regulator for terminating murine naive pluripotency [29], underlining the crucial role of METTL3 in m^6^A RNA modification. METTL14 and METTL3 are close homologues, and they always work together as METTL3-14 complexes. This complex subsequently interacts with WTAP to regulate the activity and precise localization of m^6^A methyltransferase [30]. Additionally, other unknown proteins may be involved in this complex. By proteomics, a recent study uncovered several candidate proteins, including KIAA1429 [31], required for m^6^A RNA modification in mammals. During mammalian female development, loss of function of RBM15/RBM15B results in defective XIST-mediated gene silencing on the X chromosome [32].

#### 2.1.2. m^6^A “Erasers”—Demethylases

FTO was the first m^6^A mRNA demethylase to be discovered, confirming that m^6^A RNA modification is not unalterable but can be removed. Later investigations revealed that the dysregulation of FTO may be associated with obesity, brain malformations and growth retardation [33,34,35,36], suggesting that m^6^A RNA modification may have essential regulatory functions in these diseases. Moreover, in human cell lines, knockdown of ALKB homologue 5 (ALKBH5), which belongs to the same ALKB family of FTO, causes higher m^6^A levels [37]. However, unlike FTO, ALKBH5 catalyses direct the removal of the methyl group from m^6^A-methylated adenosine instead of indirect oxidative demethylation [38].

#### 2.1.3. m^6^A “Readers”—Binding Proteins

m^6^A RNA modification works in one of two ways [25], as follows: modifying the structure of target transcripts to restrain or trigger interactions between RNAs and proteins or being directly recognized by “readers” to initiate downstream processes. According to recent studies, proteins in the YT521-B homology (YTH) family are correlated with the modulation of mRNA fate. In the cytoplasm, YTH domain-containing 1 (YTHDF1) facilitates m^6^A mRNA translation [39], and YTH domain-containing 2 (YTHDF2) aids in the degradation of mRNAs [40]. In addition, YTH domain-containing 3 (YTHDF3) collaborates with YTHDF1 to promote mRNA translation, and it enhances YTHDF2-mediated mRNA decay [41]. YTH domain-containing 1 (YTHDC1) is engaged in alternative splicing and plays an important role in nuclear export. YTH domain-containing 2 (YTHDC2) prefers to bind to transcripts that contain m^6^A RNA to accelerate RNA translation and decay [42] as well as to decrease the amount of mRNA and improve the efficiency of translation. Eukaryotic initiation factor 3 (eIF3) has been reported to directly bind to 5′UTR m^6^A mRNA [43] independent of YTHDF1. Moreover, other m^6^A “readers” have been reported, such as insulin-like growth factor 2 mRNA-binding proteins (IGF2BPs), heterogeneous nuclear ribonucleoprotein A2B1 (HNRNPA2B1), heterogeneous nuclear ribonucleoprotein C (HNRNPC) and heterogeneous nuclear ribonucleoprotein G (HNRNPG). As more m^6^A “readers” are discovered, the knowledge of m^6^A RNA modification will be more comprehensive.

### 2.2. Biofunctions of m^6^A RNA Modification

Due to breakthroughs in transcriptome sequencing of m^6^A over the past few decades, the vital roles of m^6^A RNA modification have been verified in a number of essential cellular events, including the fate of mRNA, the maintenance of stemness and even tumorigenesis [44]. The “writers”, “erasers” and “readers” involved in m^6^A RNA modification have also been linked to certain diseases [45,46,47], such as infertility, obesity, dysplasia and even cancer. Given its ubiquity in mRNA and lncRNA, further regulatory roles for m^6^A RNA modification as well as the discovery of additional m^6^A-related proteins are expected.

#### 2.2.1. Effects on mRNA Fate—Splicing, Processing, Translation and Degradation

For mRNA, m^6^A regulates the processing and alternative splicing of precursor mRNA (pre-mRNA) in the nucleus as well as the translation, degradation and decay of mRNA in the cytosol (Figure 2). Processing alternative splicing and pre-mRNA to form accurate RNA in posttranscriptional regulation allows eukaryotic organisms with limited gene counts to perform a variety of functions. METTL3, METTL14, WTAP and ALKBH5 [48] are mainly located in nuclear speckles where pre-mRNA processing occurs, implying that m^6^A RNA modification is involved in pre-mRNA processing. HNRNPC, HNRNPG and HNRNPA2B1 are associated with mRNA structure switching, thus regulating gene expression. A recent study has suggested that m^6^A RNA modification affects the binding between HNRNPC and mRNA [49], thus regulating the processing and alternative splicing of pre-mRNA. Further research has indicated that in HEK-293T cells, knockdown of *Mettl3/14* comodulates the expression of more than 5000 genes with knockdown of *Hnrnpc*, and nearly one in five of these genes may contain an m^6^A-mediated structural remodelling switch. In addition, researchers have found that the depletion of FTO enhances the m^6^A level and thus promotes the affinity of serine- and arginine-rich splicing factor 2 (SRSF2) binding with RNA, thereby increasing the number of target exons included. These data provide strong evidence of the essential role of m^6^A RNA modification during mRNA splicing.

As proteins are executors of biological functions, their synthesis and translation processes are vital to life and are regulated by various factors [37], and m^6^A is one of the factors because the presence of m^6^A in exons and surrounding stop codon regions may have an impact on protein production. *Mettl3* ablation significantly enhances translation efficiency in mouse embryonic stem cells (mESCs) and embryoid bodies (EBs) [29], indicating that m^6^A may play a translational regulatory role. Additionally, another m^6^A “reader”, YTHDF1, has been reported to increase translational output by interacting with initiation factors and ribosomes [38], providing direct evidence for the roles of m^6^A in RNA translation. Identifying more m^6^A “readers” will aid in better understanding the regulatory role of m^6^A in the translation process [50].

The flexible and rapid degradation of mRNA is critical for the response of living organisms to changeable environments. Knockdown of *Mettl3/14* modestly increases the stability of target mRNAs in mESCs [51], demonstrating that mRNA instability is associated with m^6^A RNA modification. RNA degradation has been verified to be enhanced by YTHDF2 through the transportation of mRNA for degradation [52], revealing the role of m^6^A RNA modification in mRNA degradation. The same conclusions have been reached in *Hnrnpc* [47] and *Alkbh5* [36] knockdown studies.

#### 2.2.2. Biological Consequences of m^6^A—Dysregulation in Cellular Processes and Diseases

Recent studies have revealed that m^6^A RNA is important in both physiological and pathological circumstances, including cellular stress, differentiation, development, immune response and tumorigenesis (Table 1).

##### Cellular Stress

Cellular stress includes oxidative stress, hypoxia, metabolic stress, heat shock, DNA damage ER stress and autophagy [53]. Overexpression of *Mettl3* leads to more reactive oxygen species (ROS) production through the *Keap1*–*Nrf2* antioxidant pathway in mouse renal tubular epithelial cells (mRTECs) [54]. A recent study has identified that in normal human hepatocytes, FTO triggers oxidative stress through m^6^A demethylation at the 3′UTR of peroxisome proliferator-activated receptor-γ coactivator-1α (PGC1α), an important modulator of mitochondrial metabolism, which increases the stability and enhances translation of PGC1α mRNA, thereby causing increased production of ROS [55]. YTHDF1 has been discovered to be a negative regulator in the *Keap1*–*Nrf2* pathway, as *its* knockdown induces the overexpression of *NRF2* and increases the production of antioxidants in human bronchial epithelium (Beas-2B) cells [56]. In hepatocellular carcinoma cells (HCCs), downregulation of METTL3 inhibits glycolytic capacity through the mTOR signalling pathway [57]. Depletion of *Ythdf2* causes heat shock stress by increasing HSP90, HSP60 and HSPB1 mRNA expression [58,59]. As an m^6^A “writer”, METTL16 has been confirmed to be recruited to DNA damage sites after 20–30 min of UV exposure and to methylate RNAs transported to DNA damage sites, which may be related to the nucleotide excision repair (NER) mechanism [60,61]. Additionally, FTO is correlated with the stability of *HSP70* and other genes related to ER stress through the NF-κB pathway in osteoblasts [62]. Moreover, FTO was reported to be related to autophagy by modulating the m^6^A level of Atg5 and Atg7, which was recognized by YTHDF2 [63]. FTO also participates in the demethylation of the transcripts of ULK1, thus promoting autophagy by the upregulated protein abundance of ULK1 [64].

##### Hematopoietic Development

m^6^A RNA modification plays an important role in haematopoietic and progenitor stem cell (HSPC) differentiation as well as in haematopoietic development in vertebrates. Depletion of *Mettl3* prevents double-stranded RNAs (dsRNAs) from being synthesized and activates MDA5 and RIG-I [65], ultimately causing haematopoietic stem cell (HSC) development to fail. Moreover, by elevating m^6^A levels, METTL3 stimulates c-MYC, PTEN and BCL2 mRNA translation, which inhibits the proliferation of HSPCs [66]. By degrading mRNAs involved in the Wnt signalling pathway, such as *Ccnd1*, *c-Myc* and *Axin2*, YTHDF2 suppresses this signalling pathway, thereby inhibiting the proliferation and differentiation of HSCs [67]. In HSPCs, METTL3 and YTHDF2 work together to suppress *Notch1a* [68], providing compelling evidence that m^6^A RNA modification governs haematopoietic stem cell-directed differentiation.

##### Neurogenesis

m^6^A RNA modification regulates the development of the central nervous and reproductive systems. First, through the JAK/STAT and PI3K/AKT signalling pathways, YTHDF2 [69] and FTO [70] increase the self-renewal and proliferation of neural stem/progenitor cells (NSCs). By speeding up the cell cycle of radial glial cells (RGCs), METTL3 and METTL14 promote cortical biogenesis [71]. YTHDF1 promotes synaptic transmission and transcription of LTP-related target genes in neurons, which regulates learning and memory [72].

##### Fertilization

In addition, METTL3 [73] and ALKBH5 [37,74] are involved in the modulation of motility and proliferation of sperm cells by regulating m^6^A levels. YTHDC1 [75] and YTHDF2 [76] promote oocyte proliferation and maturation by regulating the maturation and translation of CPSF6 and Trcp5 transcripts, respectively. Abnormal embryo development thus may result in spontaneous abortion (SA), which is related to the imbalance of redox reaction. Research about SA patients showed that FTO is downregulated, causing immune tolerance and angiogenesis at the placenta, thus leading to spontaneous abortion [77]. Knockdown of ALKBH5 can promote trophoblast invasion by regulating the mRNA stability of *CYR61*, thus resulting in spontaneous abortion [78].

##### Immune Response

In the immune response, research has shown that m^6^A protects RNA from being recognized by TLR3 [79] and TLR7 [80] as antigens for degradation. Additional evidence suggests that m^6^A modification plays an indispensable role in the immune response by influencing related factors, such as methylating a specific mRNA. A recent study has found an m^6^A score is useful to measure immune features of tumorigenesis in clinicopathology, such as inflammatory stages [81]. METTL3 has been shown to be responsible for the stability of *Socs* mRNA and thus regulate the TLR4/NF-κB pathway, which is related to inflammation [82]. By inactivating the TLR4/NF-κB pathway, deletion of *Mettl14* in dendritic cells (DCs) severely impairs B cell development, which affects the processes of IL-7-induced pro-B cell proliferation and causes abnormalities in gene expression critical for B cell development [83], confirming its important role in the immune response. In addition, ALKBH5 has been found to be highly expressed in spleens and lungs [70], which contain a large number of immune cells and are frequently involved in immunological reactions. Additionally, a recent study has shown that HNRNPA2B1 and HNRNPC may be associated with the abnormal immune response in endometriosis [84].

##### Cancer

Recent studies have shown a strong correlation of m^6^A RNA modification and cancer [85], including glioblastoma (GBM) and acute myeloid leukaemia (AML). In GBM, m^6^A regulates the proliferation, invasion and migration of tumours as well as the maintenance of the stemness of cancer stem cells by targeting several oncogenes [86]. By decreasing m^6^A levels of ADAM19 and increasing its expression in glioblastoma stem cells (GSCs), downregulation of METTL3/14 stimulates the proliferation and self-renewal of GSCs, ultimately resulting in GBM [87]. During the development of AML, abnormal m^6^A RNA modification of oncogenes and tumour suppressor genes plays an important role in many stages of tumorigenesis, such as proliferation and differentiation of cancer cells as well as self-renewal and cellular death of cancer stem cells [88,89,90]. Further research has revealed that AML cells have higher levels of METTL3 expression than those of normal haematological cells [91]. CEBPZ recruits METTL3 to the SP1 promoter area to increase m^6^A levels and induce translation, and SP1 subsequently activates c-MYC, a notorious oncogene, resulting in the formation of AML. Moreover, another study has found that a decreased copy number of *Alkbh5* is common in AML and has been linked to TP53 mutations, which predict a poor prognosis for AML patients [92].

##### Diabetes

Lately, the significant role of m^6^A modification in diabetes has been confirmed. In 2019, β-cell-specific *Mettl14* knockout mice were used to mimic the features of human type 2 diabetes (T2D). Some transcripts involved in the regulation of cell cycle progression and the secretion of insulin are hypomethylated and were detected by m^6^A sequencing. This study highlights the importance of RNA methylation in regulating metabolism of glucose and provides a potential therapeutic targeting of m^6^A modulators in type 2 diabetes [93]. In HepG2 cells, the expression of FTO is upregulated by high glucose, and with its upregulation, mRNA expression levels of FOXO1, G6PC and DGAT2 were significantly increased [94].

**Table 1 nanomaterials-12-00389-t001:** Roles of m^6^A RNA methylation and dysregulation in cellular processes and diseases.

Processes /Diseases	m^6^A Regulator	Cells/ Organisms	Effect of Gene Loss/Gain of Function	Mechanism	Ref.
	METTL3	mRTECs	↑ROS	↑METTL3/*Keap1*/*Nrf2*	[54]
	FTO	L02 cells	↓ROS	↓FTO/↑PGC-1α	[55]
	YTHDF1	Beas-2B cells	↓Hypoxia adaptation	↓YTHDF1/*Keap1*/*Nrf2*-AKR1C1	[56]
Cell stress	METTL3	HCCs	↓Glycolytic capacity	↓METTL3/mTORC	[57]
	YTHDF2	MEFs	↑Heat shock stress	↓YTHDF2/↑HSP90, HSP60, HSPB1	[58,59]
	METTL16	MEFs	↑DNA damage	↑METTL16/γH2AX	[60,61]
	FTO	Mice	↑ER stress	↓FTO/↓HSP70/↑NF-κB	[62]
	FTO	Mice	↑autophagy	↓FTO/↓Atg5, Atg7	[63]
	FTO	293T cells	↑autophagy	↓FTO/↑ULK1	[64]
Haematopoietic	METTL3	HSCs	↓Proliferation, ↓differentiation	↓METTL3/↑MDA5/RIG-I	[65]
development	METTL3	HSPCs	↑Differentiation, ↓cell proliferation	↑METTL3/c-MYC/BCL2/PTEN	[66]
	YTHDF2	HSCs	↑Regeneration	↓YTHDF2/↑*Wnt* target genes	[67]
	METTL3	HSPCs	↑Endothelial to haematopoietic transition	↓METTL3/↑YTHDF2/↓*Notch1a*	[68]
Neurogenesis	YTHDF2	NSPCs	↓Self-renewal	↓YTHDF1/JAK–STAT	[69]
	FTO	NSCs	↓Proliferation, ↓differentiation	↓FTO/BDNF/PI3K/*Akt2*/*Akt3*	[70]
	METTL3/14	RGCs	↑Neurogenesis, ↑cell cycle	↓METTL3/14/↑*Neurog2*/*Neurod1*	[71]
	YTHDF1	Mice	↓Learning, memory defects	↓YTHDF1/*Camk2a*	[72]
Fertilization	METTL3	Zebrafish	↓Sperm motility	↓METTL14/11-KT/17β-E2	[73]
	ALKBH5	Mice	↓Fertility	↓ALKBH5/↑*Dnmt1*	[37,74]
	YTHDC1	Germ cells	↓Oocyte growth, maturation	↓YTHDC1/CPSF6/SRSF3	[75]
	YTHDF2	Mice	↓Oocyte maturation	↓YTHDF2/Trpc5	[76]
	FTO	SA patients	↑Spontaneous abortion	↓FTO/VEGFA, VEGFR	[77]
	ALKBH5	SA patients	↑Spontaneous abortion	↓FTO/↑CYR61	[78]
Immune	METTL3	Mice	↓T cell proliferation	↓METTL3/IL-7/STAT5/SOCS	[82]
response	METTL14	DCs	↓B cell development	↓METTL14/TLR4/NF-κB	[83]
Cancer	METTL3/14	GBM	↑Proliferation and self-renewal of GSCs	↓METTL3/14/↑ADAM19	[87]
	METTL3	AML	↓Cell cycle and differentiation of leukaemic cells	↓METTL3/↑c-MYC	[91]
	ALKBH5	AML	↓Prognosis of AML patients	↓ALKBH5/↑TP53	[92]
Diabetes	METTL14	Mice	↓Insulin secretion	↓METTL14/IGF1–AKT–PDX1	[93]
	FTO	HepG2 cells	↓Glucose metabolism	↓FTO/FOXO1/G6PC/DGAT2	[94]

### 2.3. Detection

The direct detection of m^6^A bases is difficult because the base pairing properties remain unchanged and cannot be distinguished from regular bases by reverse transcription. Methylated RNA immunoprecipitation followed by high-throughput sequencing (MeRIP-seq), also known as m^6^A RNA immunoprecipitation sequencing (m^6^A-seq), has been widely used to detect m^6^A at a resolution of 110–200 nt since it was developed in 2012 [95].

In the past decade, several approaches with the resolution of a single base, including photo crosslinking-assisted m^6^A sequencing (PA-m^6^A-seq) [96], m^6^A individual nucleotide resolution cross-linking and immunoprecipitation sequencing (miCLIP-seq) [97], site-specific cleavage, and radioactive labelling, followed by ligation-assisted extraction and thin-layer chromatography (SCARLET) [98], have been developed (Table 2). A recent study has introduced an efficient method to directly detect m^6^A bases by m^6^A-sensitive RNA endoribonuclease-facilitated sequencing (m^6^A-REF-seq) [99] with higher sensitivity and a lower false-positive rate, eliminating the dependence of traditional identification methods on antibodies.

Moreover, several methods have been developed to detect the total amount of m^6^A in RNA [100], including colorimetry, m^6^A dot blot analysis and high-performance liquid chromatography–tandem mass spectrometry (HPLC–MS/MS). Furthermore, directly detecting the effect of changing any site and in any organism is promising with the advancement of genome engineering.

## 3. m^6^A RNA Modification Modulates the Bioimpacts of Nanoparticles

NPs have been widely used in production and life due to their distinct physicochemical features, which is accompanied with concerns over their bioimpacts and mechanisms involved. Therefore, there is a need to study and comprehend the impacts in terms of the cellular and molecular mechanisms of the bioimpacts of NPs. The mechanisms involved are partly clarified, including oxidative stress, cytotoxicity, neurotoxicity and genotoxicity [101], which depend on physiochemical features, such as the size and crystallinity of nanoparticles.

However, there is more evidence suggesting the possibility of NP-induced genetic toxicity. DNA methylation, histone modifications and RNA modifications as well as many other epigenetic mechanisms have been proposed and widely studied in recent years. As more research has been performed and more knowledge is available, the roles of epigenetics in the biological function of NPs are gaining increasing attention. Most notably, epigenetic dysregulation has been linked to clinical disorders, including cancer and neurological diseases. Thus, it is possible that NP-mediated epigenetic modulations can also contribute to pathogenesis to a certain level. One of the characteristics of cancer is global DNA hypomethylation, which can be passed down from generation to generation [102]. For instance, the levels of pri-miRNA-1275 are increased after Ag-NP exposure in neural stem/progenitor cells derived from human embryonic stem cells (hESCs) with obvious downregulation of their target genes, ADAMTS9 and SHANK2, which encode proteins involved in axonal guidance signalling and are linked to brain damage and neurodegenerative diseases [103]. The link between nanoparticles and m^6^A RNA modification is less well understood, but evidence is beginning to emerge in the past two years. Therefore, we will introduce these studies and summarize the role of m^6^A RNA modification in the bioimpacts induced by NPs.

Recently, we established a 3D high-throughput screening system using constructed kidney organoids to study the nephrotoxicity of black phosphorus quantum dots (BPQDs) [104]. Interestingly, we found significant endoplasmic reticulum (ER) stress and insulin sensitivity induced by BPQDs in the kidney. Furthermore, considering their easy inhalation ability, we used lung cells to further evaluate the bioimpacts of BPQDs and their correlation with m^6^A RNA modification. Surprisingly, the global m^6^A level was increased after BPQD exposure in lung cells, and after the protein expression of m^6^A-related proteins was examined, we further clarified the significant role of ALKBH5 in this event; the expression of “writers” and other “erasers” did not change, but ALKBH5 decreased with BPQD exposure [105]. Transcriptome and epitranscriptome data have provided evidence of the correlation between the abnormal elevation of total m^6^A levels and the aberrant expression of genes related to ferroptosis involving ER stress, iron homeostasis and iron homeostasis, which has also been verified by evidence of iron overload, lipid peroxidation, glutathione peroxidase 4 (GPX4) downregulation and glutathione (GSH) depletion in vitro. Moreover, the mechanism of the downregulation of antiferroptosis-related genes is that YTHDF2, a m^6^A “reader”, recognizes and promotes the decay of mitochondrial homeostasis-containing and antilipid peroxidation genes modulated by m^6^A, which has been verified by m^6^A RNA immunoprecipitation (RIP)-qPCR in lung cells. Our study reported the roles of m^6^A RNA modification in BPQD-induced ferroptosis and for the first time revealed the mechanism involved, thereby providing a promising biomarker and a drug target for BPQD-induced ferroptosis (Figure 3).

Carbon black nanoparticles (CBNPs) have been reported to have a close correlation with many diseases, some of which are involved in reproductive systems and abnormal behaviours. A recent study has shown the relationship between CBNP-induced pulmonary fibrosis and m^6^A RNA modification [106]. The processing of pri-miRNA-126 is a DiGeorge syndrome critical region gene 8 (DGCR8)-dependent process that regulates the downstream PI3K/AKT/mTOR pathway. This study showed that CB exposure causes pulmonary fibrosis and activates the PI3K–AKT–mTOR pathway by decreasing the m^6^A level of pri-miRNA-126 and its binding with DGCR8. Additionally, another study has reported that CBNP exposure during pregnancy influences maternal behaviours and partially causes abnormal neurobehaviors and the development of the reproductive system in offspring, all of which are linked to m^6^A RNA modification [107]. This study revealed the potential correlation between the abnormal maternal behaviours induced by CBNPs and the decreased m^6^A level. Further studies need to be performed to provide more information about this intriguing phenomenon.

Moreover, a previous study on the bioimpacts of high dots of multiwalled carbon nanotubes (MWCNTs) on *Arabidopsis thaliana* [108] has suggested the potential correlation of m^6^A RNA modification and the bioimpacts of MECNTs (Figure 4). In detail, the growth of *Arabidopsis thaliana* is severely inhibited after MWCNT exposure, presenting abnormal behaviours of roots and leaves as well as suppression of auxin and photosynthesis signalling. MWCNTs also cause oxidative stress, thus activating the antioxidant mechanism. Combined data from m^6^A-seq and RNA-seq have shown that m^6^A RNA modification may negatively regulate the transcription of related genes after MWCNT exposure, including plant hormone transduction signalling pathways and protein phosphorylation. For the first time, these findings offer insight into the molecular pathways underlying MWCNT phytotoxicity and plant defensive responses to MWCNTs. In addition, many studies have shown that cellular metabolism can be reprogrammed through m^6^A RNA modification during viral infection. Recently, a study uncovered that metal–protein nanoparticles (MPNPs) in macrophages infected by vesicular stomatitis virus (VSVs) polarize macrophages and stimulate immunological responses related to METTL14 [109]. Further research has revealed that a high dose of interferon-beta (IFN-β) increases the expression of METTL14, an anti-VSV protein. The same conclusions have been reached for influenza viruses (H1N1(WSN)), which are also negative-sense single-stranded RNA viruses. Overall, these findings shed light on the antiviral role of METTL14 and suggest that manipulating METTL14 may be a viable technique for combating additional negative-sense single-stranded RNA virus infections.

In 2021, research related to TiO_2_-NPs and m^6^A RNA modification was reported [110]. In this study, researchers aimed to determine whether mitochondrial phospholipid hydroperoxide glutathione peroxidase (mPHGPx) can sustain cardiovascular function and bioenergetics in offspring after direct exposure to TiO_2_-NPs by inhalation during pregnancy in female mice and to reveal an epitranscriptomic mechanism that contributes to this phenomenon. The results showed that in offspring, the enzymatic function of mPHGPx is considerably decreased, and the m^6^A levels are increased. These findings attribute the abnormally high m^6^A level of mPHGPx to reduced antioxidant capacity and the resulting mitochondrial and cardiac impairments that last into adulthood after gestational nTiO_2_ exposure through inhalation.

In addition to those studies related to bioimpacts of NPs induced by m^6^A RNA modification, there is some research about electrochemical immunosensors for m^6^A detection based on nanotechnology. Silver nanoparticles and SiO_2_ nanospheres are used in the electrochemical immunosensor, providing a promising detection platform for m^6^A [111]. Moreover, nanoparticles have emerged as promising carriers in cancer therapy by delivering drugs to target. A recent study showed that N^6^-methyladenosine (m^6^A) mediated by METTL3 upregulated long noncoding RNA LINC00958, which subsequently promoted HCC progression through the miR-3619-5p/HDGF axis [112]. Therefore, it is promising that RNA m^6^A modification functions as a robust tool to modify RNA used in cancer therapy, as the veil of RNA m^6^A modification in the bioimpacts of NPs is lifted completely.

## 4. Conclusions and Outlook

Despite a large number of findings connected to the functional roles of epigenetics in the bioimpacts of NPs [23], many vital gaps still have to be filled, particularly in the area of RNA m^6^A modification. Epigenetic modifications can be utilized to study nanoparticle toxicity and, more crucially, to predict their toxicity in preclinical research, according to recent evidence from both experimental and epidemiological studies. A link between nanotoxicity and epigenetic changes following exposure to specific nanoparticles has been demonstrated in many experimental studies in vivo and in vitro [113,114,115]. YTHDC1 and METTL14, in particular, have been identified as possible endometrial carcinoma diagnostic and prognostic indicators [116]. Therefore, RNA m^6^A modifications may function as biomarkers not only to detect and predict the toxicity of NPs but also to evaluate the process of diseases triggered by NPs as the role of RNA m^6^A modification in the bioimpacts of NPs is further understood.

The roles of DNA methylation and histone acetylation in the biological functions induced by NPs have been reported in great detail [23,25,117]. However, information on the roles of RNA m^6^A modification is lacking. Moreover, most studies focus on the bioeffects of one type of NP on RNA m^6^A modification, whereas clarifying the process involved in RNA m^6^A modification and NPs in greater depth could provide broader insights into the understanding of nanomaterials. The precise impacts of the physicochemical features of NPs, including their size and surface charge, in modulating RNA m^6^A modification are also unknown to some extent.

The genetic effects of NP-induced bioimpacts related to RNA m^6^A modification are another gap that must be filled. Studies involving longer research periods are needed to illustrate the consequences of RNA m^6^A modification correlated with NPs and to determine whether these biological effects can be passed from generation to generation [118]. Several model organisms, such as *Danio rerio,* can serve as efficient in vivo models for determining the potential epigenetic mechanisms in the toxicity triggered by NPs and the genetic effects of the bioimpacts of NPs in this field [119,120]. Zebrafish are low-cost and simple to feed in the lab, and they can be used to efficiently test agents through multiple routes of exposure, including direct exposure to water, which is especially significant for the toxicology of environmental pollution. Because of the short life cycle and distinct developmental stages, transgenerational studies can be completed in a relatively short period of time. Furthermore, specific physiological effects can be studied at diverse stages of the whole development of zebrafish.

Finally, dysregulation of RNA m^6^A modification has been linked to some pathological diseases, such as obesity, as well as several mental disorders, such as neurodegenerative diseases, and even cancer [16,42]. Recent studies have found that NPs are closely related to dysregulation in some cellular processes and diseases when exposed to the body through the digestive system, respiratory system and skin tissue [121]. Thus, it is possible that NP-induced RNA m^6^A modification can similarly lead to some diseases to a certain extent. Therefore, addressing the roles of RNA m^6^A modification in the bioimpacts of NPs and the involved mechanisms as well as using various practical in vivo and in vitro models are important challenges for the future.

Last but not least, with the remarkable development of RNA nanotechnology, RNA nanoparticles can be used as carriers in drug delivery systems [122]. In order to improve targeting efficiency and achieve optimal therapeutic efficacy, physiochemical features can be modulated on purpose. Disease diagnosis and healthcare based on RNA nanotechnology can achieve early diagnosis, early detection and early treatment, lowering disease preventive costs and improving human health protection. Furthermore, RNA nanotechnology has a wide range of potential applications in disease prevention based on vaccine creation. With more investigation, it is promising to target m^6^A-methylation-based epitranscriptomics using the nanoparticle as an “epigenetic drug” for cancer therapy.

## Figures and Tables

**Figure 1 nanomaterials-12-00389-f001:**
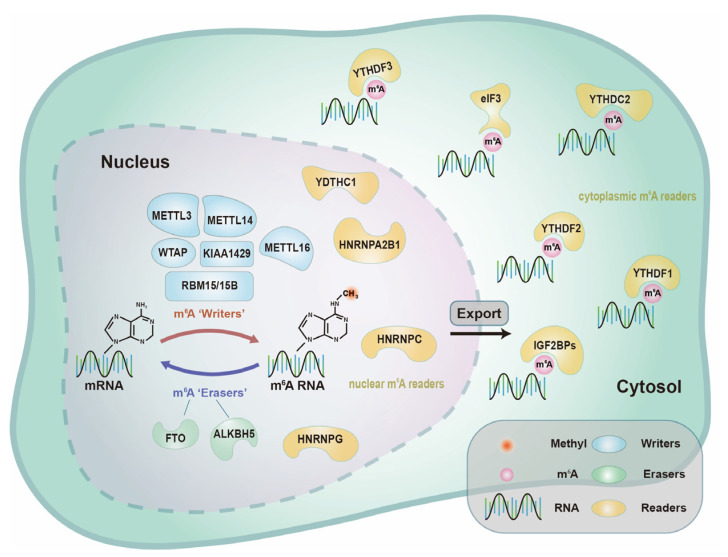
The molecular composition of m^6^A RNA methylation. “Writers” of m^6^A are methyltransferases that add methyl groups to mRNA, including METTL3, METTL14, WTAP, KIAA1429 METTL16 and RBM15/15B. “Erasers” of m^6^A are demethylases that remove methyl groups from m^6^A RNA, including FTO and ALKBH5. “Readers” of m^6^A recognize m^6^A RNA and thus trigger downstream events, including nuclear m^6^A “readers” (such as YTHDC1, HNRNPA2B1, HNRNPC and HNRNPG) and cytoplasmic m^6^A “readers” (such as YTHDF1, YTHDF2, YTHDF3, YTHDC2, IGF2BPs and eIF3).

**Figure 2 nanomaterials-12-00389-f002:**
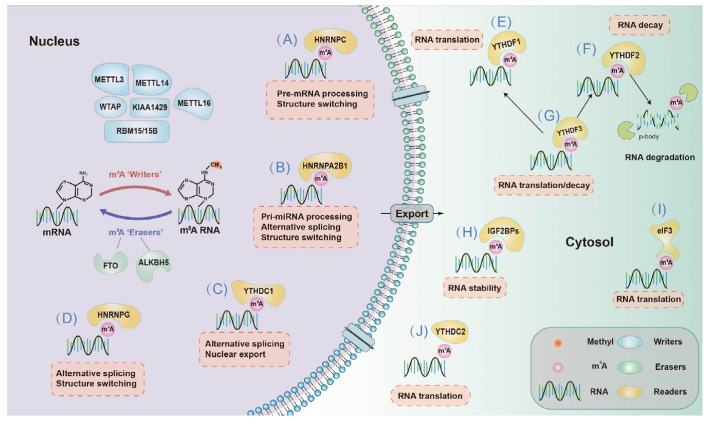
m^6^A RNA modification influences mRNA fate. (**A**) HNRNPC is involved in the processing of pre-mRNA and structural switching. (**B**) HNRNPA2B1 plays an important role in pri-miRNA processing, alternative splicing and structural switching of mRNA. (**C**) YTHDC1 is engaged in alternative splicing and nuclear export. (**D**) HNRNPG participates in alternative splicing and mRNA structure switching. (**E**) YTHDF1 facilitates mRNA translation. (**F**) YTHDF2 aids in the decay of mRNAs. (**G**) YTHDF3 collaborates with YTHDF1 to promote mRNA translation, and it enhances YTHDF2-mediated mRNA decay. (**H**) IGF2BPs containing IGF2BP1/2/3 enhance RNA stability. (**I**) eIF3 is engaged in RNA translation. (**J**) YTHDC2 accelerates RNA translation.

**Figure 3 nanomaterials-12-00389-f003:**
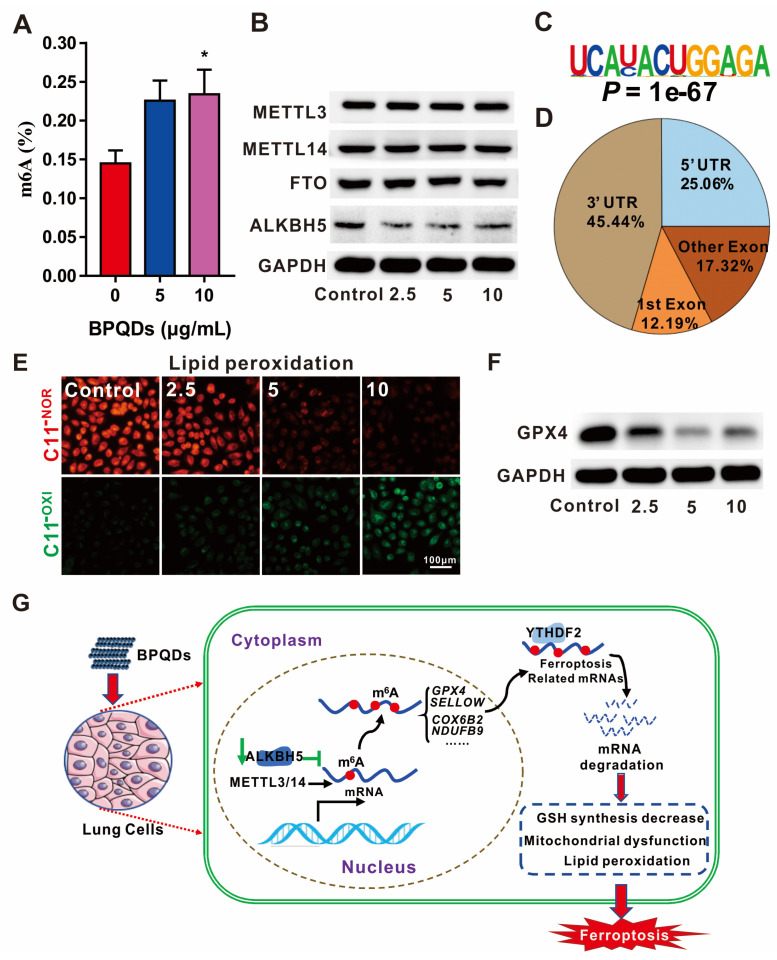
The roles of m^6^A RNA modification in ferroptosis induced by BPQDs. (**A**) The global m^6^A level is increased after BPQD exposure in a concentration-dependent manner. (**B**) The expression of m^6^A-related proteins, as detected by Western blotting, suggests that BPQDs decrease the expression of the demethylase ALKBH5 rather than FTO or the METTL3/14 methylases. (**C**) Most significant m^6^A-binding motif. (**D**) m^6^A peaks are mainly located at 3′ and 5′ UTR of mRNA. (**E**) Fluorescent images showing the lipid peroxidation level as measured with C11 BODIPY staining. Scale bar = 100 µm. (**F**) The expression of GPX4 protein is decreased after BPQD exposure with GAPDH as a loading control. (**G**) BPQDs decrease ALKBH5 expression and thus lower the m^6^A level of ferroptosis-related mRNAs recognized by the YTHDF2 m^6^A “reader”, promoting mRNA degradation. * *p* < 0.05. * compared to no treatment. As a result, GSH synthesis decreases and mitochondrial dysfunction, lipoperoxidation and ferroptosis occur. Reproduced from [97] with permission from *Small Methods*.

**Figure 4 nanomaterials-12-00389-f004:**
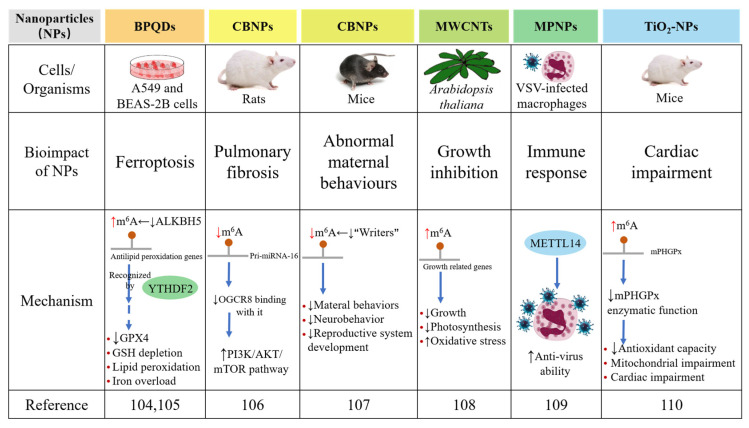
Roles of m^6^A RNA modification in the bioimpacts of nanoparticles and the mechanisms involved. BPQDs induce ferroptosis by upregulating the total m^6^A level in A549 and BEAS-2B cells. CBNPs induce pulmonary fibrosis by downregulating the total m^6^A level in rats and also cause abnormal maternal behaviours by downregulating the total m^6^A level in mice. MWCNTs cause growth inhibition by upregulating the total m^6^A level in *Arabidopsis thaliana*. MPNPs improve anti-virus ability in VSV-infected macrophages. TiO_2_-NPs cause cardiac impairment by upregulating the total m^6^A level [104,105,106,107,108,109,110].

**Table 2 nanomaterials-12-00389-t002:** Methods for detecting m^6^A residues. (Extremely large: ≥300 μg; Large: 300 ng < n < 300 μg; Little: ≤300 ng).

Methods	Resolution	Sample RNA Demand (n)	Need for Antibodies	References
MeRIP-seq	100–200 nt	Extremely large	Yes	[95]
PA-m^6^A-seq	20–30 nt	Large	Yes	[96]
miCLIP-seq	Single base	Large	Yes	[97]
SCARLET	Single base	Large	No	[98]
m^6^A-REF-seq	Single base	Little	No	[99]
Colorimetry	Total amount of m^6^A	Little	Yes	[100]
m^6^A dot-blot		Large	Yes	
HPLC–MS/MS		Large	No	

## Data Availability

Not applicable.

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
