# Peer review of "Nanoparticle-Induced m6A RNA Modification: Detection Methods, Mechanisms and Applications"

_nanomaterials, 2022, doi:10.3390/nano12030389_

Round 1
Reviewer 1 Report
The manuscript "The roles of m6A RNA modification in the bioimpacts of nanoparticles and mechanisms involved" by Yi Wang et al. is devoted to the role of m6A RNA modification in nanoparticle bioimpacts and the mechanisms involved in this process. Understanding the transformation of nanoparticles will optimize their structure to reduce negative effects. The review is written on a rather relevant topic, since the use of nanoparticles is a promising direction that is widely studied and used. The text of the review is logically structured, systematized, interesting to read and can be accept after minor revision:
- Line 6 “A With the increasing application”- a very strange phrase.
- Line 10 “these events has attracted substantial research intersect” - a very strange phrase.
- It is unclear for what purpose the authors of the review use Figure 3, which is only an enumeration of the results described in the article Ruan, F.; Zeng, J.; Yin, H.; Jiang, S.; Cao, X.; Zheng, N.; Han, C.; Zhang, C.; Zuo, Z.; He, C. RNA m6A Modification Alteration by Black Phosphorus Quantum Dots Regulates Cell Ferroptosis: Implications for Nanotoxicological Assessment. Small Methods 2021, 5, doi:10.1002/smtd.202001045.
- The visual appearance of Figure 4 looks more like a table than a drawing. Perhaps it makes sense to call this figure a table, or change the drawing.
- Figure 5 repeats Figure 4 in meaning.
- Phrases “others...” in figure 5 does not carry any semantic information, and also does not have an explanation in the text of the manuscript. Perhaps they should be removed from the picture.
- Authors should pay attention to the use of the terms “in vitro” and “in vivo” unchanged with italics (lines 36, 320, 398, 417, 433).
- Lines 116, 119, 122, 124-127, 133, 201, 314-315, 321, 352 when writing the words “writers”, “erasers” and “readers”, authors use quotation marks. It is necessary to bring the words into a single form and in other cases on the lines 58, 70, 75-76, 93, 102, 105, 162, 165.
- It is important to bring the use of the abbreviation m6A in the entire text of the manuscript into a single form.
- Authors should pay attention to keywords and more thoughtfully compile a list of effective keywords for a more competent and organic description of the text of the manuscript and attract a much larger number of interested readers.
Author Response
Reviewer #1
General comments:
The manuscript "The roles of m6A RNA modification in the bioimpacts of nanoparticles and mechanisms involved" by Yi Wang et al. is devoted to the role of m6A RNA modification in nanoparticle bioimpacts and the mechanisms involved in this process. Understanding the transformation of nanoparticles will optimize their structure to reduce negative effects. The review is written on a rather relevant topic, since the use of nanoparticles is a promising direction that is widely studied and used. The text of the review is logically structured, systematized, interesting to read and can be accept after minor revision.
Response:
We appreciate Reviewer #1 for the positive evaluation and valuable comments on our manuscript. The link between nanoparticles and m6A RNA modification is now less well understood, but evidence is beginning to emerge in the past two years. Therefore, a summary of nanoparticle-induced m6A RNA modification from current research is quite timely and significant. This review will be helpful for the understanding of the relationship between m6A RNA modification and nanoparticles and may provide a possible way to facilitate the design of nanoparticles by deliberately tuning m6A RNA modification applied to identify causative toxic substances accumulated in biological samples in the future. The specific comments and suggestions from Reviewer #1 were very helpful in clarifying our manuscript.
Specific comments:
- Line 6 “A With the increasing application”- a very strange phrase.
Response: We are sorry for this mistake in our original manuscript and we have corrected it.
Revision:(Line 6) With the increasing application of nanoparticles (NPs) in medical and consumer items, it is quite needed to ensure their safety.
- Line 10 “these events has attracted substantial research intersect” - a very strange phrase.
Response: We are sorry for this mistake in our original manuscript and we have corrected it.
Revision: As m6A (N6-methyladenosine) RNA modification is one of the most prevalent RNA modifications involved in many diseases and essential biological processes, the relationship between nanoparticles and m6A RNA modification for the modulation of (Line 10) these events has attracted substantial research interest.
- It is unclear for what purpose the authors of the review use Figure 3, which is only an enumeration of the results described in the article Ruan, F.; Zeng, J.; Yin, H.; Jiang, S.; Cao, X.; Zheng, N.; Han, C.; Zhang, C.; Zuo, Z.; He, C. RNA m6A Modification Alteration by Black Phosphorus Quantum Dots Regulates Cell Ferroptosis: Implications for Nanotoxicological Assessment. Small Methods 2021, 5, doi:10.1002/smtd.202001045.
Response: We shall clarify the purpose to use Figure 3 here. For one thing, Figure 3 contains some significant results from the article Ruan, et al. RNA m6A Modification Alteration by Black Phosphorus Quantum Dots Regulates Cell Ferroptosis: Implications for Nanotoxicological Assessment. Small Methods 2021, 5, doi:10.1002/smtd.202001045. This article was the first one to report the relationship between BPQDs and RNA m6A modification. For another, considering the expression and presentation style, we do believe using words together with illustration is much better than only words. Therefore, we cited those important data to make our introduction to this part more direct and vivid. All in all, we use Figure 3 in this review and we would appreciate it very much if you could consider the meaning of using Figure 3 again.
- The visual appearance of Figure 4 looks more like a table than a drawing. Perhaps it makes sense to call this figure a table, or change the drawing.
Response: Thank you for your suggestion and after careful consideration, we have changed Figure 4 into Table 3.
Revision:
- Figure 5 repeats Figure 4 in meaning.
Response: Thank you for your advice and we have decided to change Figure 5 into a Graph Abstract (GA).
Revision:
- Phrases “others...” in figure 5 does not carry any semantic information, and also does not have an explanation in the text of the manuscript. Perhaps they should be removed from the picture.
Response: Thanks for your suggestion and we have considered this question when writing this assay. we shall clarify that “others” contains hematopoietic development, immune response, neurogenesis, fertilization, cancer and so on, which we have introduced in the text in detail. Besides, we have changed Figure 5 to GA as a summary of this review. Therefore, we would appreciate it very much if you could consider the meaning of using this expression.
- Authors should pay attention to the use of the terms “in vitro” and “in vivo” unchanged with italics (lines 36, 320, 398, 417, 433).
Response: We are so sorry for these mistakes and we have corrected them all in the manuscript.
Revision:(Line 36) Just because of the widespread use of NPs, their biological effects on organisms and mechanisms involved have been extensively studied over the past few decades, using many models both in vitro and in vivo.
(Line 320) Data of transcriptome and epitranscriptome provided an evidence to the correlation between the abnormal elevation of total m6A level and the aberrant expression of genes related to ferroptosis involving ER stress, iron homeostasis, iron homeostasis, which was also verified by evidences of iron overload, lipid peroxidation, the downregulation of glutathione peroxidase 4 (GPX4) and glutathione (GSH) depletion in vitro.
(Line 398) A link between nanotoxicity and epigenetic changes following exposure to specific nanoparticles has been demonstrated in plenty of experimental studies in vivo and in vitro.
(Line 417) Some splendid model organisms such as Danio rerio can serve as an efficient model in vivo for figuring out the potential epigenetic mechanisms in the toxicity triggered by NPs and the genetic effects of the bioimpacts of NPs in this filed.
(Line 433) Therefore, addressing the roles of RNA m6A modification in the bioimpacts of NPs and mechanisms involved, using various and practical models both in vivo and in vitro are important challenges for the future.
- Lines 116, 119, 122, 124-127, 133, 201, 314315, 321, 352 when writing the words “writers”, “erasers” and “readers”, authors use quotation marks. It is necessary to bring the words into a single form and in other cases on the lines 58, 70, 75-76, 93, 102, 105, 162, 165.
Response: We are so sorry for these mistakes and we have corrected them all with quotation marks in the manuscript.
- It is important to bring the use of the abbreviation m6A in the entire text of the manuscript into a single form.
Response: We are so sorry for these mistakes and we have corrected them all in the manuscript.
- Authors should pay attention to keywords and more thoughtfully compile a list of effective keywords for a more competent and organic description of the text of the manuscript and attract a much larger number of interested readers.
Response: Thanks for your advice and we decide to change keywords from “RNA m6A; nanoparticles; mechanism; bioimpacts” into “nanosafety; (N6-methyladenosine) RNA modification; function of m6A; epigenetic”. If you have any better idea about our keywords, please contact us and we would appreciate it very much.
Additional corrections:
- Manuscript has been improved by a native English speaker.
- Typo and/or grammatical errors were corrected while revising the manuscript.

Reviewer 2 Report
Wang et al. comprehensively reviewed the function of (N6-methyladenosine) RNA modification in the bioimpacts of nanoparticles and the underlying mechanism. The subject is interesting and has great significance.
- In my idea, the manuscript can be strengthened by adding some information related to RNA nanotechnology and its advancements in biomedical fields.
- Page 1, lines 36 and 38. The authors mentioned “On the one hand” two times in a row. It better be reworded to avoid redundancy.
- In the Introduction, the authors need to elaborate on the role of nanoformulations as theranostic tools in the fight against different diseases by citing and briefly discussing the following papers one by one; DOI: 1016/j.sbsr.2021.100417, DOI: 10.3390/polym13183153
- In section 2.2.2., I advise subcategorizing the biological consequences of m6A-dysregulation in cellular processes and diseases. i.e., for stem-cell differentiation (2.2.2.1.), etc. Also, a schematic representation is highly encouraged.
- All the figures lack quality and must be replaced with more transparent ones.
- Page 7, lines 259-260. The sentence is vague and should be reworded.
- Table 1 should be placed before section 2.3.
- What about the role of m6A modification in diabetes, autophagy, Spontaneous Abortion, etc.? Table 1 and section 2.2.2. must be improved.
- In figure 4, name of the lung cancer cell line is missed.
Reviewer 3 Report
The manuscript nanomaterials-1547886 is a review article aiming at elucidating the link between nanoparticles (NPs) and N6-methyladenosine (m6A) RNA modification, which might help understanding the role of NPs in the development of pathological conditions. Due to the growing scientific interest in NPs, the paper might be of interest for the journal’s readers. However, the paper presents with important issues, as follows.
The level of English is quite poor and seriously affects the readability of the manuscript. There are many issues with verb tenses (some phrases include present and past tenses). There are unclear sentences. Therefore, an extensive revision of the language is needed prior to publication.
Most of the paper presents general background or methodological information, being the paragraph describing NPs and m6A methylation the shortest one (100 lines approximately). I believe that this should be the real focus of the review, and I encourage the authors to rewrite and expand this paragraph providing more detailed and well referenced information on this core topic.
I appreciate the use of Figures, but I believe that Figures 4 and 5 overlap, so I recommend rearranging the information contained in them in a more meaningful way.
Under “Conclusions and outlook”, instead of summarizing previous knowledge, it would be better to emphasize current gaps that need to be filled and potential applications of such information.
Round 2
Reviewer 2 Report
It is acceptable now.
Reviewer 3 Report
The quality of the revised manuscript has increase substantially, making it suitable for publication